

# A novel autoencoder approach to feature extraction with linear separability for high-dimensional data

Jian Zheng[1], Hongchun Qu[1,2], Zhaoni Li[1], Lin Li[1], Xiaoming Tang[2] and Fei Guo[2]

[1] College of Computer Science and Technology, Chongqing University of Post and Telecommunications, Chongqing, China
[2] College of Automation, Chongqing University of Posts and Telecommunications, Chongqing, China

## ABSTRACT

Feature extraction often needs to rely on sufficient information of the input data, however, the distribution of the data upon a high-dimensional space is too sparse to provide sufficient information for feature extraction. Furthermore, high dimensionality of the data also creates trouble for the searching of those features scattered in subspaces. As such, it is a tricky task for feature extraction from the data upon a high-dimensional space. To address this issue, this article proposes a novel autoencoder method using Mahalanobis distance metric of rescaling transformation. The key idea of the method is that by implementing Mahalanobis distance metric of rescaling transformation, the difference between the reconstructed distribution and the original distribution can be reduced, so as to improve the ability of feature extraction to the autoencoder. Results show that the proposed approach wins the state-of-the-art methods in terms of both the accuracy of feature extraction and the linear separabilities of the extracted features. We indicate that distance metric-based methods are more suitable for extracting those features with linear separabilities from high-dimensional data than feature selection-based methods. In a high-dimensional space, evaluating feature similarity is relatively easier than evaluating feature importance, so that distance metric methods by evaluating feature similarity gain advantages over feature selection methods by assessing feature importance for feature extraction, while evaluating feature importance is more computationally efficient than evaluating feature similarity.

## INTRODUCTION

High-dimensional data usually contains rich features, through extracting the important features, those irrelevant attributes in high-dimensional data can be filtered, thereby achieving data dimensionality reduction (*Xue, Zhang & Browne, 2015*). Hence, feature extraction is considered to be one of the important methods for data dimension reduction (*Bo, Kay & He, 2016*).

Feature extraction is a hot topic in recent years, aiming to gain the most valuable features from the input data (*Tao, Hou & Nie, 2016*; *Luo, Nie & Chang, 2018*). High dimensionality of data, the so-called the curse of dimensionality, brings negative effects for feature

Corresponding author
Hongchun Qu, hcchyu@gmail.com

extraction (*Gui, Sun & Ji, 2018*; *Chakraborty & Pal, 2015*). Upon a low-dimensional space, those relations between the data are relatively compact but they may become sparse upon a high-dimensional space (*Bing, Liao & Zhou, 2021*), *e.g.*, the data space with more than 10 dimensionalities (*Zhou, Kumar & Hou, 2011*). Clearly, sparse relations between data are usually considered to be an unfavorite factor for feature extraction since feature extraction needs to rely on the relations between data (*Bo, Wan & Bi, 2021*). Beyond that, those latent features scattered in subspaces inside a high-dimensional space not only inspect the ability of methods to extract features (*Wang, Wang & Chang, 2016*), but also test their extraction efficiency. Hence, it is a challenge for feature extraction from high-dimensional data.

Recently, some opinions have been proposed for feature extraction, for instance, distance metric-based methods, where, the typical representative is the well-known Mahalanobis distance-based methods, which evaluates the similarity between samples using the covariance matrix of data (*De Maesschalck, Jouan-Rimbaud & Massart, 2000*). Furthermore, *Ying, Wen & Shi (2018)* proposed the intrinsic semi-supervised metric learning (ISSML) based on a distance metric for feature extraction. Similarly, the methods implemented in (*Zadeh, Hosseini & Sra, 2016*; *Luo, 2017*) also applied distance metrics. Certainly, also including, the information-theoretic metric learning is (ITML) (*Mei, Liu & Karimi, 2014*) employed a distance metric to obtain features. These methods (*Ying, Wen & Shi, 2018*; *Zadeh, Hosseini & Sra, 2016*; *Luo, 2017*; *Mei, Liu & Karimi, 2014*) address the issues of symmetric positive-definite matrix minimization during feature extraction, but there are several problems in them, (1) since most of them use iterative calculation while performing feature selection, optimization issues have to be addressed iteratively. (2) Most of them need to rely on parameter selection to obtain those desired features. Usually, feature selection-based methods are also considered to be used for feature extraction. Such methods achieve feature extraction through analyzing the information of feature subsets, for example, the cheap feature selection method based on $k$-means algorithm (*Marco, Pérez & Lozano, 2021*) selects the $m$ features with the highest relevance measure through obtaining a clustering for each subset of features. Although the method (*Marco, Pérez & Lozano, 2021*) is a novel measurement for feature relevance, which is beneficial for feature selection, however, calculating per subset of features needs to spend a lot time cost. In order to reduce the correlation between features, some measurements for quickly assessing features are proposed, *e.g.*, the information entropy metric (*Pham, Siarry & Oulhadj, 2019*), whereas the method (*Pham, Siarry & Oulhadj, 2019*) has a bias toward features, which may result in appearing selecting deviation during feature extraction. Another kind of feature selection method depends on eigen decomposition, such as, locally linear embedding (LLE) (*Hettiarachchi & Peters, 2015*; *Akpudo & Hur, 2020*), multi-manifold discriminant isometric feature mapping (MMD-ISOMAP) (*Bo, Xiang & Zhang, 2016*), ISOMAP-KL (*Neto & Levada, 2020*), however, they cannot assess the importance of the features in the background space explicitly.

Neural network-based methods are favored because of excellent feature capture ability (*Qu, Zheng & Tang, 2022*), *e.g.*, Multilayer Perceptron Neural Network (*Sun, Huang & Wong, 2017*). For dimension reduction, feature extraction and data compression, autoencoder-based networks provide an interpretable approach for the unknown

meaningful insights (*Ang, Mirzal & Haron, 2016*) by learning non-identity mapping functions (*Zheng et al., 2022*), for instance, *Al-Hmouz, Pedrycz & Balamash (2022)* developed interpretable data representation for data dimensionality reduction using logic-oriented and granular logic autoencoders, and such as, autoencoder (*Majumdar, 2019*) for image compression, and blind denoising autoencoder (*Yang, Herranz & Van de Weijer, 2020*) for denoising. In addition, sparse autoencoders are used as an unsupervised feature extractor to serve data dimensionality reduction, feature extraction and data mining (*Wan, He & Tang, 2018*), *e.g.*, *Chen, Hu & He (2018)* proposed sparse autoencoder (SAE) for feature extraction of ferroresonance overvoltage waveforms in power distribution systems. *Yan & Han (2018)* used stacked sparse autoencoder (SSAE) to extract effective features. In addition, the autoencoders (*Qu et al., 2021*; *Qu, Zheng & Tang, 2022*; *Zheng et al., 2022*) also successfully capture the low-dimensional features from high-dimensional data, however, these captured low-dimensional features do not show good linear separability. In terms of addressing high-dimensional complex problems, deep methods are the state-of-the-art solution in many disciplines (*Abadía-Heredia et al., 2022*), *e.g.*, video and language processing, etc.

In this study, our motivation is to extract the features with linear separabilities from the data in a high-dimensional space. Thus, we proposed a novel autoencoder method based on Mahalanobis distance metric of rescaling transformation. The proposed method does not have to address any optimization issue, and also it can focus on the whole data distribution.

We summarize the main contributions of this work as follows:

(i) Distance metric-based methods are more suitable for extracting those features with linear separabilities from high-dimensional data than feature selection-based methods.

(ii) Assessing feature similarity in a high-dimensional space is relatively easier than evaluating feature importance, therefore, distance metric approaches by evaluating feature similarity have more advantages than feature selection approaches by evaluating feature importance in terms of feature extraction.

(iii) The computational time of distance metric-based algorithms is higher than that of feature selection-based algorithms upon a high-dimensional space.

This paper is organized as follows. Section 2 describes the proposed method and implements the proposed model, including training for the model and parameter configuration. Experiment datasets, competing methods, and experiment description are given in Section 3. Section 4 presents experiment results. Section 5 draws conclusions.

## METHODS

### Theory

Given a sample $X = \{ x_i \mid 1 \leq i \leq N \}$, and $X \subseteq \Re^d$. $\Re^d$ is the $d$-dimensional Euclidean space. $P$ is the probability distribution of $X$, denoted as original probability distribution. $u(X)$ and $\Gamma_x$ are the mean vector and the covariance matrix of $X$, respectively. Let us assume that $Z = \{ z_j \mid 1 \leq j \leq N \}$ is the reconstructed $X$, and $Z \subseteq \Re^d$. $Q$ is the probability distribution of $Z$, denoted as approximate probability distribution. Similar, $u(Z)$ and $\Gamma_z$ are the mean

vector and the covariance matrix of $Z$, respectively. The K–L divergence (*Tao et al., 2009*) between the two distributions $P$ and $Q$ is given in Eq. (1).

$$K(P||Q) = \frac{1}{2}\left[\log|\Gamma_z| - \log|\Gamma_x| + \text{tr}(\Gamma_z^{-1}\Gamma_x) + \text{tr}(\Gamma_z^{-1}D_{xz})\right]. \tag{1}$$

where $|\Gamma| = \det(\Gamma)$. The $\text{tr}(\cdot)$ is the trace of a matrix. $D_{xz} = (u(X) - u(Z))(u(X) - u(Z))^T$ is a symmetrical matrix. Training a distance metric is equivalent to finding a rescaling of a sample which replaces each $x_i$ with $M^T x_i$ (*Feng, Wang & Jin, 2019*), so the K-L divergence in Eq. (1) can be converted into Eq. (2), having

$$K_L^*(P||Q) = \frac{1}{2}\left[\log|M^T\Gamma_z M| - \log|M^T\Gamma_x M| + \text{tr}\left((M^T\Gamma_z M)^{-1}(M^T(\Gamma_x + D_{xz})M)\right)\right]. \tag{2}$$

where $M$ is a metric matrix and satisfies $A* = MM^T$, and $M \in \Re^{d \times d_0}, d_0 \leq d$. The K-L divergence in Eq. (2) is the rescaling transformation for the K-L divergence in Eq. (1) using the distance metric matrix $A^\star$. To reduce the difference between the approximate distribution Q and the original distribution P, we consider Mahalanobis distance metric for K-L divergence in Eq. (2), having

$$K - L(d_{A*}) = K_L^*(P||Q) + \sum_{1 \leq i,j} d_{A*}(x_i, z_j). \tag{3}$$

$d_{A*}(x_i, z_j)$ is Mahalanobis distance between $x_i$ and $z_j$ using $A^\star$. The advantage of doing this is that the Mahalanobis distance using $A^\star$ can appropriately measure similarities between the input sample and the reconstructed input sample because of non-negativity (i.e., $d_{A*}(x_i, z_j) \geq 0$), distinguishability (i.e., $d_{A*}(x_i, z_j) = 0 \Leftrightarrow x_i = z_j$) and symmetry (i.e., $d_{A*}(x_i, z_j) = d_{A*}(z_j, x_i)$) (*Feng, Wang & Jin, 2019*). Equation (4) gives the calculation of $d_{A*}(x_i, z_j)$, where $A^\star$ can be decomposed as $A* = MM^T$.

$$d_{A*}(x_i, z_j) = \sqrt{(x_i - z_j)^T A * (x_i - z_j)}. \tag{4}$$

## Model implementation

A classic auto encoder (AE) consists of an input layer, a hidden-layer and an output layer. For AE, the loss error is often measured by using the distance between the original input instance, the predicted instances, and the reconstructed instance (*Theis, Shi & Cunningham, 2017*). Typically, using divergence metrics or expanding autoencoder structures (*e.g.,* enlarging the number of hidden layers) is more helpful for autoencoders to characterize the data distribution and to learn the desired representations (*Lu, Cheng & Xiao, 2017*). As such, we designed an autoencoder with multiple-hidden layers, namely m-AE, and $m \geq 1$, as shown in Fig. 1. In addition, the K–L divergence in Eq. (3) was used to increase the ability of $m$-AE to capture low-dimensional feature representations. The loss error $\nabla_{KL}(\mathbf{w}, \mathbf{b})$ in m-AE is given as follows:

$$\nabla_{KL}(\mathbf{w}, \mathbf{b}) = \sum ||e_X - e_Z||^2 + K - L(d_{A*}). \tag{5}$$

where $e_X$, $e_Z$ are the inputting and the reconstructed inputting, respectively. $\nabla_{KL}(\mathbf{w}, \mathbf{b})$ isupdated through using the backpropagation manner.

To better train the proposed model, we carefully studied part hyper parameters in the model. For the rest of hyper parameters, their default values were used.

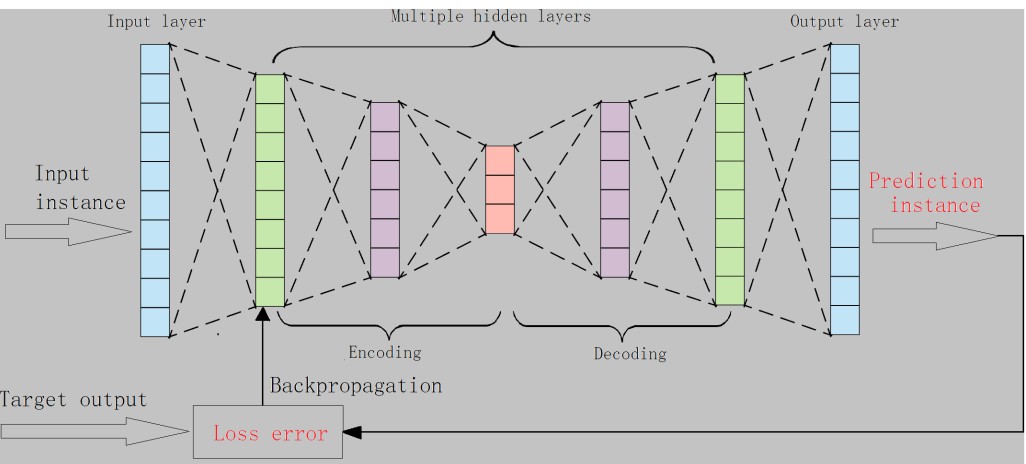

**Figure 1** The structure of the proposed m-AE.

(i) Optimizer. Common optimizers are Adam, RMSprop, SGD, Momentum, Nesterov, etc. However, we selected Adam as the optimizer of m-AE, since Adam has the ability to handle sparse gradients (*Kingma & Ba, 2015*). Compared with other optimizers, Adam is more suitable for high-dimensional data. Moreover, Adam can provide different adaptive learning rates for different hyper parameters.

(ii) Activation function. Gradient vanishing is easily to be induced during passing gradients backwards for neural networks, in this case, the probability of gradient vanishing caused by activation function Sigmoid is relatively high. Similar to Sigmoid, activation function tanh also suffers from this problem. While for activation function ReLu, the phenomenon of gradient vanishing is partially alleviated, meaning that gradient vanishing does not appear in the positive interval of ReLu. Furthermore, ReLu converges much faster than Sigmoid and Tanh. Therefore, we chose ReLu as the activation function of m-AE.

(iii) Iteration epoch. We dynamically adjust the iteration epoch according to training accuracy. For instance, when training accuracy starts to change from large to small, we reduce iteration epoch in order to prevent over-fitting. When the difference in accuracy between training and testing is minimal, the current iteration epoch can be accepted and training procedure is stopped.

We give the training algorithm for m-AE in Algorithm 1. In the algorithm, the training set *Train_set* is divided into two datasets $T^{Cro\_train}$, $T^{Cro\_val}$ in step 1. Since m-AE has multiple hidden layers, we set $m$ in the range of $O_m$, in order to determine the $m$, the dataset $T^{Cro\_train}$ is used to train m-AE. The data set $T^{Cro\_val}$ is used for the validation of the network structure of m-AE. To get the optimal $m$, denoted as $m_{opt}$, the cross-validation is implemented in step 2 to step 18, where the procedure of step 6 and step 10 describes the calculation process of loss error $\nabla_{KL}(\mathbf{w}, \mathbf{b})$. After gaining the optimal $m$, m-AE is trained using the training set *Train_set*. Using backpropagation manner updates network parameters until m-AE can converge, as shown in step 18 to step 28. The procedure shown

in step 29 to step 33 indicates that the maximum training accuracy *Train_acc* are outputted and the well trained m-AE is saved.

**Algorithm 1.** Training for m-AE.

Input: Training set *Train_set*, $A* = I \in \Re^{d \times d}$ is an identity matrix, iteration epoch *T*, *L*, parameter $O_m$.

Output: Training accuracy *Train_acc*.

**Begin**

1. *Train_set* is divided into $T^{Cro\_train}$, $T^{Cro\_val}$;

2. **for** $t = 1$ **to** $T$ **do:**

3.     **foreach** $m$ **in** $O_m$:

4.         Decompose $A^\star$ as satisfying $A* = MM^T$ using eigen decomposition.

5.         Calculate loss error $\nabla_{KL}(\mathbf{w}, \mathbf{b})$ using Eq. (5) and the procedure is summarized as following:

6.         The procedure:

7.                 Calculate $K_L^*(P||Q)$ using Eq. (2).

8.                 Calculate $d_{A*}(x_i, z_j)$ using Eq. (4).

9.                 Take $K_L^*(P||Q)$ and $d_{A*}(x_i, z_j)$ into Eq. (3) to calculate $K - L(d_{A*})$.

10.                 For any $x_i$, $x_j$, calculate $\nabla_{KL}(\mathbf{w}, \mathbf{b})$ using Eq. (5).

11.         Calculate training accuracy $T\_acc = m - AE(T^{Cro\_train}; m; t)$;

12.         Validate m-AE using data set $T^{Cro\_val}$;

13.         Calculate validation accuracy $V\_acc = m - AE(T^{Cro\_val}; m; t)$

14.         Update weight $\mathbf{w} \leftarrow \mathbf{w} + \nabla \mathbf{w}$.

15.         Update $A^\star$ as $MM^T$.

16.         Until $A^\star$ and hyper parameters converge.

17.     **end foreach**

18. **end for**

19. Get the optimal value of $m$, *i.e.*, $m_{opt} = \arg\max(V\_acc)$;

20. **for** $l = 1$ **to** $L$ **do:**

21.         Decompose $A^\star$ as satisfying $A* = MM^T$ using eigen decomposition.

22.         Train m-AE using training set Train_set and $m_{opt}$;

23.         Update network parameters using the optimizer Adam;

24.         Calculate loss error $\nabla_{KL}(\mathbf{w}, \mathbf{b})$ using Eq. (5);

25.         Calculate training accuracy $Training\_acc(l) = m - AE(\text{Train\_set}; m_{opt})$;

26.         Update $A^\star$ as $MM^T$;

27.         Using backpropagation manner updates network parameters;

28. **end for**

29. Select the $l$ so that $l_{max} = \arg\max(Training\_acc(l))$;

30. Get the maximum training accuracy Train_acc in the $l_{max}$-th iteration;

31. *Train_acc* = m-AE(*Train_set*; $m_{opt}$, $l_{max}$);

32. Output *Train_acc*

33. Save the well trained m-AE(*Train_set*; $m_{opt}$, $l_{max}$);

**End**

**Table 1 Benchmark datasets.**

| Dataset | Data volume | Data dimensionality | features |
|---|---|---|---|
| Iris | 150 | 4 | 3 |
| Primary | 339 | 17 | 2 |
| Hepatitis | 155 | 19 | 2 |
| Dermatology | 366 | 33 | 6 |

# EXPERIMENTS

## Datasets and assessment metrics

To verify the performance of the proposed m-AE, we selected four benchmark datasets with different data dimensions from the UCI machine learning repository (*Blake & Merz, 1998*). The attributes of the four benchmark datasets are summarized in Table 1.

Receiver operating characteristic curve (ROC) and corresponding area under curve (AUC) are usually used to assess the precision of machine learning methods. Therefore, AUC is taken as the assessment metric of method precision.

## Competing and benchmark methods

Since m-AE applies the distance metric of rescaling transformation, the methods based on a distance metric were used for comparisons, including ISSML (*Ying, Wen & Shi, 2018*) and ITML (*Mei, Liu & Karimi, 2014*). Certainly, the method based on feature selection was also considered, *i.e.*, MMD-ISOMAP (*Bo, Xiang & Zhang, 2016*). In addition, autoencoder-based approaches were used as a comparison, *e.g.*, the SAE (*Chen, Hu & He, 2018*). Furthermore, to further examine the effects of the distance metric of rescaling transformation on the performance of m-AE, a benchmark model was developed with m-AE as a reference. The developed benchmark model used the same structure and parameter configuration of m-AE without using the distance metric of rescaling transformation, namely AE-BK.

We implemented the corresponding algorithms of the six models using Python on Tensorflow framework. While for those parameters of competing methods, we adopted those values observed in the corresponding literature. Certainly, unless otherwise stated, the five corresponding algorithms all run on the same GPU and apply the same experimental configuration settings.

## Experiment description

Experiments were conducted on the four benchmark datasets in order to validate the ability of these six models to extract features and their efficiency.

**Experiment I**. To test the robustness of m-AE. The proposed m-AE has multiple hidden layers, since the number of hidden layers (*i.e.*, the m) significantly affects the precision of feature extraction, the *m* needs to be firstly verified, *i.e.*, robustness testing of the model, let m set in the range of {1, 2, 3, 4, 5, 7, 10, 15, 20}.

**Experiment II**. To test the ability of feature extraction for the six models. The six models were run on the four benchmark datasets, and then the testing results were analyzed.

 

**Experiment III**. To compare the efficiency of our method with competing methods. These methods were performed on four benchmark datasets and observed their running time.

**Ablation experiments**. To verify that using the distance metric of rescaling transformation can be beneficial for extracting linearly separable features, the ablation experiments were also designed.

In addition, to eliminate randomness during the experiment and present an objective result, we used cross-validation to verify the six models. We randomly selected two datasets from the four benchmark datasets as the training set to train the six models. Once the six models were well trained, they were tested on the four benchmark datasets, respectively. The process was repeated five times, independently, then we took the average of five testing results was as a measurement.

## RESULTS

### Experiments on robustness

Results in Fig. 2 show that the performance of the proposed m-AE and the benchmark model AE-BK improves along with increasing of $m$, and then the performance remains stable when $m$ reaches a certain size, *i.e.*, $m = 3$. This means that m-AE and AE-BK are not sensitive to large $m$ on the four benchmark datasets, *i.e.*, their network structures are robust within a reasonable range. Therefore, let $m$ be equal to three in subsequent experiments.

### Comparisons of accuracy extraction

Results in Table 2 show that the proposed m-AE wins the four competing models and the benchmark model in the accuracy of feature extraction on all considered instances. For competitors, ISSML, ITML and SAE outperform MMD-ISOMAP in most benchmark datasets for the extracted accuracy.

### Comparisons of linear separability

The results of ablation experiments in Fig. 3A show that compared with the models without using distance metrics, *e.g.*, AE-BK, SAE, the models using distance metrics (including m-AE, ISSML, ITML) perform much better on most datasets in the extracted accuracy of the features with linear separabilities. Similar, the models using distance metrics also win the model using feature selection, as shown in Fig. 3B. To observe the linear separabilities of the extracted features from the four benchmark datasets, we projected these extracted features onto two-dimensional space, and then visualized them. Figure 4 displays the results of visualized distribution on the four benchmark datasets by the six models. The visualized results show that it is optimal for the separation distance between different types of features extracted by m-AE, meaning that compared with competing and benchmark models, m-AE is a winner in terms of the linear separabilities of the extracted features. Together, these results imply that distance metric-based methods have advantages over feature selection-based methods in terms of extracting the features with linear separabilities.

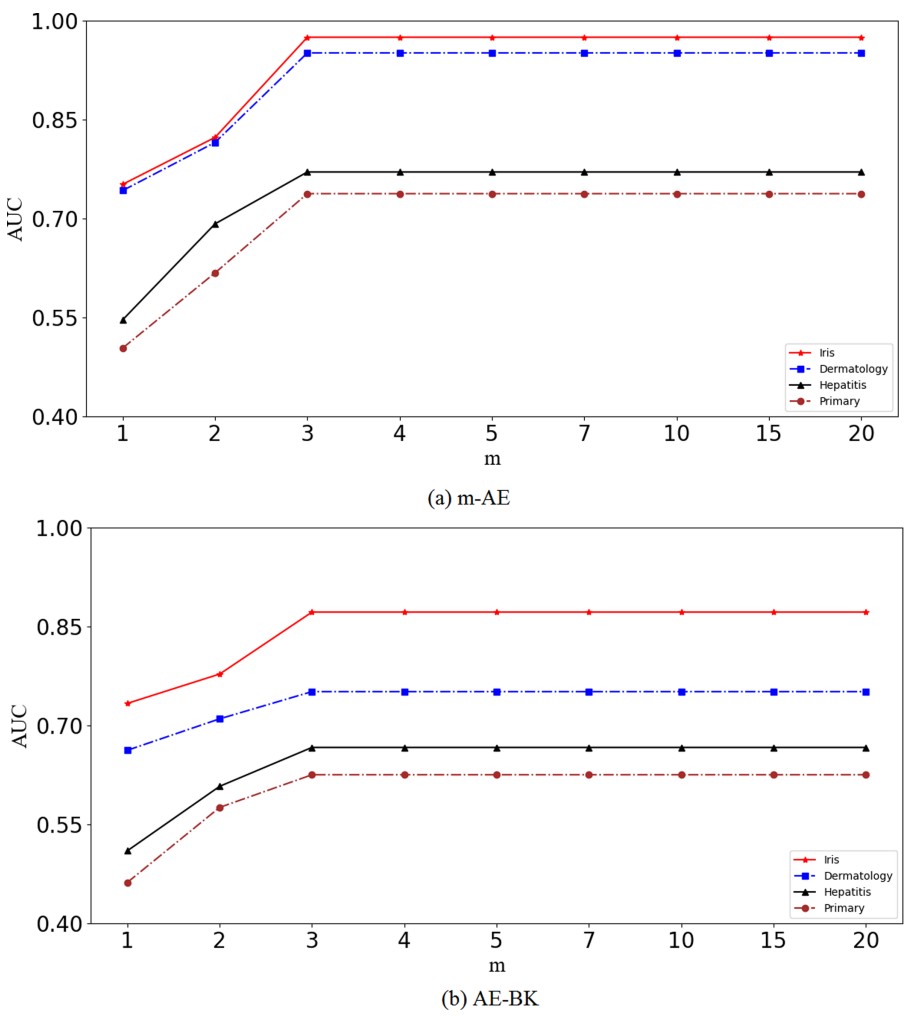

(a) m-AE

(b) AE-BK

**Figure 2** **Validation of robustness.**

**Table 2** **Accuracy of feature extraction.** The best accuracy for each dataset is shown in bold. The models using a distance metric are marked as the symbol √. The models using feature selection are marked as the symbol ≠. The models without both a distance metric and feature selection are marked as the symbol ×.

|  | Iris | Dermatology | Hepatitis | Primary |
|---|---|---|---|---|
| m-AE (√) | **0.9744** ± 0.0157 | **0.9506** ± 0.0137 | **0.7703** ± 0.0753 | **0.7375** ± 0.0534 |
| ISSML (√) | 0.9402 ± 0.0154 | 0.8931 ± 0.0284 | 0.7131 ± 0.0642 | 0.6886 ± 0.0865 |
| ITML (√) | 0.9488 ± 0.0120 | 0.9374 ± 0.0246 | 0.7457 ± 0.0622 | 0.6816 ± 0.0745 |
| MMD-ISOMAP (≠) | 0.9247 ± 0.0053 | 0.7680 ± 0.0377 | 0.6897 ± 0.0657 | 0.6664 ± 0.0733 |
| SAE (×) | 0.9571 ± 0.0227 | 0.8707 ± 0.0892 | 0.6773 ± 0.0373 | 0.6700 ± 0.0166 |
| AE-BK (×) | 0.8715 ± 0.1533 | 0.7511 ± 0.0099 | 0.6666 ± 0.0771 | 0.6252 ± 0.1052 |

## Running time

Figure 5 displays the running time of methods. Obviously, the advantage of m-AE in running time is not as significant as that in both the extracted accuracy and the linear

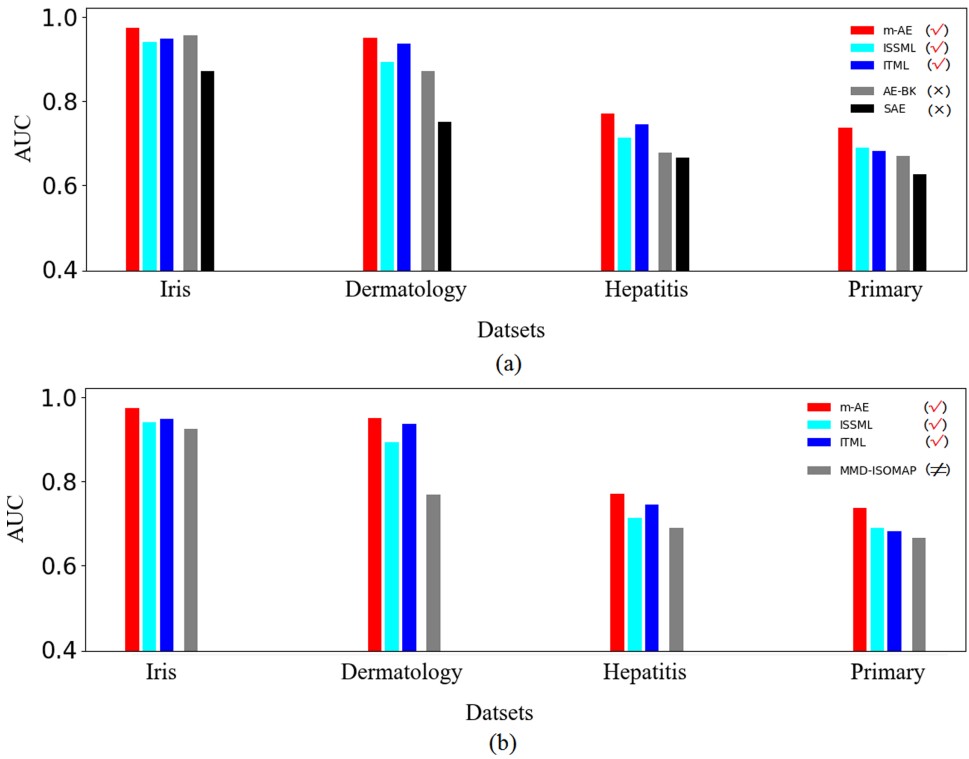

**Figure 3 Results of ablation experiments.** (A) Comparisons between using distance metrics and without using distance metrics. These models using distance metrics are marked as the symbol √. The models without both distance metrics and feature selection are marked as the symbol ×. (B) Comparisons between using distance metrics and using feature selection. These models using feature selection are marked as the symbol ≠.

separabilities of the extracted features. MMD-ISOMAP spends less in running time on most benchmark datasets than distance metric-based methods, meaning that the execution efficiency of feature selection-based methods is higher than that of distance metric-based methods when running upon a high-dimensional space. Distance metric-based methods take a lot of time to calculate the distance between each point pair upon a high-dimensional space, so as to increase the running time.

## DISCUSSION

### Insights gained from investigation

Compared with the competitors, the proposed m-AE has outstanding advantage in term of both the accuracy of feature extraction and the linear separabilities of the extracted features on high-dimensional data. We interpret it as following. On one hand, Mahalanobis distance in Eq. (3) can appropriately measure similarities between the input sample and the reconstructed input sample, so as to minimize the loss error of m-AE in Eq. (5). As such, m-AE gains the desired accuracy of feature extraction. On the other hand, we performed a rescaling on K-L divergence metric in Eq. (2) by using $A^\star$ in Eq. (4), which effectively allows the extracted features to present linear separabilities, because the rescaling can maximized

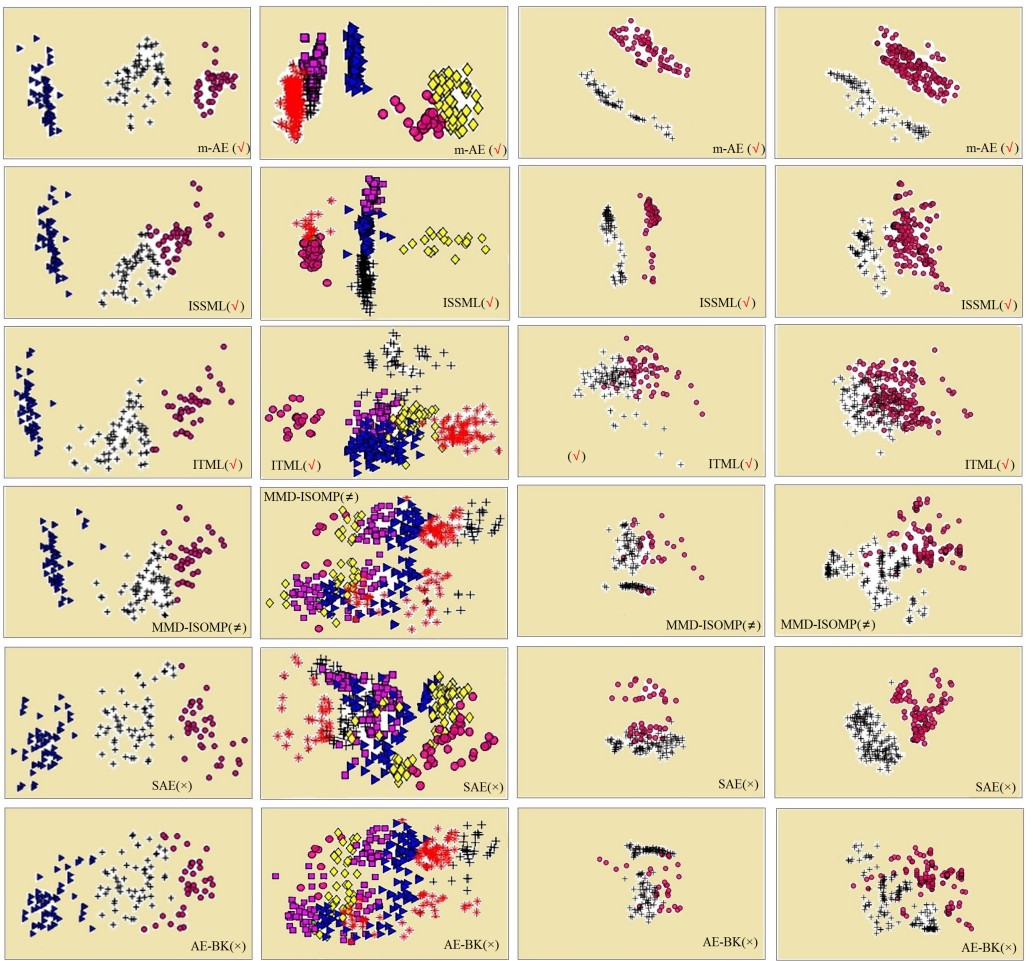

**Figure 4** **Visualization distributions.** The four datasets are Iris, Dermatology, Hepatitis, Primary from left to right, respectively. The different extracted features are marked with different shapes and colors. The models using distance metrics are marked as the symbol $\sqrt{}$. The models using feature selection are marked as the symbol $\neq$. The models without both distance metrics and feature selection are marked as the symbol $\times$.

the classification distance between the extracted different types of features. Hence, the features extracted by m-AE present linear separabilities than competitors. Overall, m-AE outperforms the competitors in extracted accuracy and the linear separabilities of the extracted features.

In a high-dimensional space, distance metric-based methods easily evaluates the feature similarity by calculating the distance between the data, however, feature selection-based methods relatively difficulty assess the feature importance. Therefore, distance metric-based methods, *e.g.*, ISSML (*Ying, Wen & Shi, 2018*) and ITML (*Mei, Liu & Karimi, 2014*), are more suitable for extracting those low-dimensional features with the linear separability from high-dimensional data than feature selection-based methods. However, the computational time of feature selection-based methods, *e.g.*, MMD-ISOMAP (*Bo, Xiang & Zhang, 2016*), is lower than that of distance metric-based methods in a high-dimensional space, since

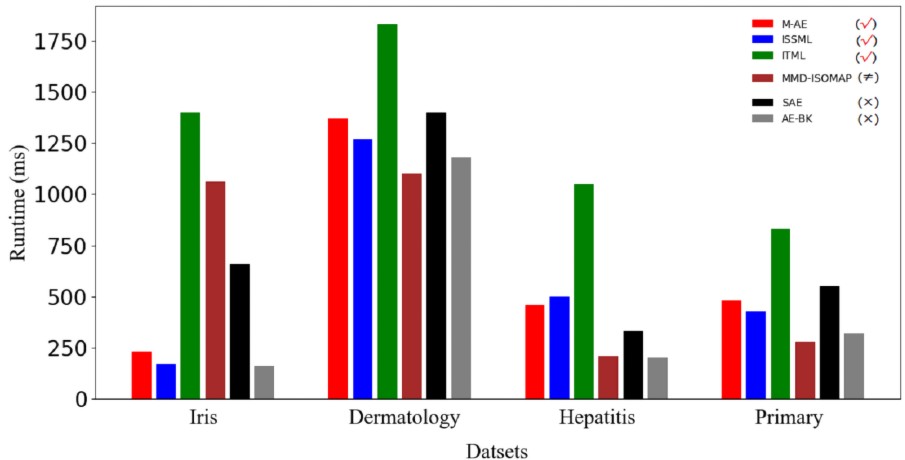

**Figure 5** **Runtime on benchmark datasets.** The models using distance metrics are marked as the symbol $\checkmark$. The models using feature selection are marked as the symbol $\neq$. The models without both distance metrics and feature selection are marked as the symbol $\times$.

distance metric-based methods spend too much in calculating the distance between each point pair.

Although autoencoders have excellent feature capture capabilities, they may perform poorly in extracting linearly separable features, *e.g.*, SAE (*Chen, Hu & He, 2018*). Whereas, this deficiency of autoencoders can be remedied by introducing a distance metric. Certainly, there are many methods of distance metrics, *e.g.*, Wasserstein distance metric (*Lei, Su & Cui, 2019*; *Zheng et al., 2022*), Bhattacharyya distance metric (*Mariucci & Reiß, 2017*).

### Limitations

The ability of the autoencoder to extract linearly separable features depends on the reconstructed data distribution, while the reconstruction of the data distribution is achieved by the Mahalanobis distance metric of rescaling transformation. Upon a high-dimensional space, the calculation of Mahalanobis distance metric is relatively complicated than that up a low-dimensional space. Moreover, matrix factorization operation needs to be implemented for each computation, therefore, the proposed model is trained using large-scale high-dimensional data until it can converge, which may take longer training epoch.

### CONCLUSIONS

This article proposed a novel autoencoder method using Mahalanobis distance metric of rescaling transformation to extract linearly separable features from the data in the high-dimensional space. The difference between the reconstructed distribution and the original distribution can be reduced by implementing Mahalanobis distance metric of rescaling transformation, so that the autoencoder can extract the desired features. Finally, results on real high-dimensional datasets show compared with competing methods, the proposed method is a winner in both the accuracy of feature extraction and the linear

separabilities of the extracted features. We find that the linear separabilities of those features obtained by the distance metric-based methods are better than that of obtained by the feature selection-based methods. Upon a high-dimensional space, since evaluating feature similarity is relatively easier than evaluating feature importance, distance metric-based methods have more advantages than feature selection-based methods for linearly separable feature extraction, however, feature selection-based methods are better than distance metric-based methods in computational efficiency. In future work, we will look at exploring low-dimensional feature extraction from high-dimensional data under noise disturbance.

### Funding

This work was supported by the National Natural Science Foundation of China (No. 61871061). The funders had no role in study design, data collection and analysis, decision to publish, or preparation of the manuscript.

### Grant Disclosures

The following grant information was disclosed by the authors:
National Natural Science Foundation of China: 61871061.

### Competing Interests

The authors declare there are no competing interests.

### Author Contributions

- Jian Zheng conceived and designed the experiments, performed the experiments, analyzed the data, performed the computation work, authored or reviewed drafts of the article, and approved the final draft.
- Hongchun Qu conceived and designed the experiments, performed the experiments, analyzed the data, authored or reviewed drafts of the article, and approved the final draft.
- Zhaoni Li analyzed the data, prepared figures and/or tables, and approved the final draft.
- Lin Li analyzed the data, prepared figures and/or tables, and approved the final draft.
- Xiaoming Tang performed the computation work, prepared figures and/or tables, and approved the final draft.
- Fei Guo performed the computation work, prepared figures and/or tables, and approved the final draft.

### Data Availability

Raw data is available in the Supplemental Files.

### Supplemental Information

Supplemental information for this article can be found online at http://dx.doi.org/10.7717/peerj-cs.1061#supplemental-information.

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
