# Peer review of "A novel autoencoder approach to feature extraction with linear separability for high-dimensional data"

_PeerJ Computer Science, doi:10.7717/peerj-cs.1061_

## Round 0.1 · original submission · Minor Revisions

Please enhance the literature review with recent papers. Although Reviewer 1 has requested that you cite some references, I do not expect you to include these citations. If you do not include them, this will not influence my decision in any sense.

Remember that it is PeerJ policy that additional references suggested during the peer-review process should only be included if the authors are in agreement that they are relevant and useful, not under the suggestion of the reviewer.

Reviewer 1 ·

Basic reporting

This work presents a novel method for feature extraction based on Mahalanobis distance defined by the covariance matrix between features. Some comments:
- Abstract should be more clear
- Separation of graphics from the sequence of the text makes it more difficult to read.
- Highlight all assumptions and limitations of your work.
- Conclusions should provide some lessons learnt.
- Related works section does not mention recent research effors in new approaches to extract meaningful features. Authors are advised to refer to the following related articles to add some discussions: [1] Supervised contrastive learning over prototype-label embeddings for network intrusion detection, Information Fusion, 2022 [3] Effective Feature Extraction via Stacked Sparse Autoencoder to Improve Intrusion Detection System, IEEE Access, 2018 [3] A predictive hybrid reduced order model based on proper orthogonal decomposition combined with deep learning architectures, Expert Systems with Applications, 2022

Experimental design

The design of experiments and the ablation strategy seem correct

Validity of the findings

The selection of different datasets and alternative models looks enough to validate the results

Reviewer 2 ·

Basic reporting

no comment

Experimental design

The experimental design of the authors can be accepted.

Validity of the findings

The findings demonstrated by the authors are very interesting. Especiallly, distance metric-based methods are more suitable for extracting those features with linear separabilities from high-dimensional data than feature selection-based methods

Additional comments

To extract feature from the data in a high-dimensional space, this paper proposed a novel autoencoder approach based on Mahalanobis distance metric of rescaling transformation. Through performing rescaling transformation on Mahalanobis distance metric, then the transformed Mahalanobis distance metric is introduced into the autoencoder, so as to improve the ability of feature extraction to the model.
The issue is very interesting, and these findings are valuable. Moreover, the experiments and the results are acceptable through verifying the source code provided by the authors. Based on the studied problem and these findings, I recommend that this paper is acceptable. In addition, to highlight the studied problem and these findings, I suggest the title “A novel autoencoder approach to feature extraction with linear separability for high-dimensional data”

Annotated reviews are not available for download in order to protect the identity of reviewers who chose to remain anonymous.

---

## Round 0.2 · accepted · Accept

Congratulations on the acceptance of your manuscript.